

# Continuously varying critical exponents in long-range quantum spin ladders

**Patrick Adelhardt[⋆] and Kai P. Schmidt[†]**

Department of Physics, Staudtstraße 7, Friedrich-Alexander-Universität
Erlangen-Nürnberg (FAU), Germany

⋆ patrick.adelhardt@fau.de , † kai.phillip.schmidt@fau.de

## Abstract

We investigate the quantum-critical behavior between the rung-singlet phase with hidden string order and the Néel phase with broken SU(2)-symmetry in quantum spin ladders with algebraically decaying unfrustrated long-range Heisenberg interactions. To this end, we determine high-order series expansions of energies and observables in the thermodynamic limit about the isolated rung-dimer limit. This is achieved by extending the method of perturbative continuous unitary transformations (pCUT) to long-range Heisenberg interactions and to the calculation of generic observables. The quantum-critical breakdown of the rung-singlet phase then allows us to determine the critical phase transition line and the entire set of critical exponents as a function of the decay exponent of the long-range interaction. We demonstrate long-range mean-field behavior as well as a non-trivial regime of continuously varying critical exponents implying the absence of deconfined criticality contrary to a recent suggestion in the literature.



# 1  Introduction

While in electromagnetism the interaction between charged particles is long-range decaying as a power-law with distance, in condensed matter systems the interaction is typically screened, justifying to consider short-range interactions in most microscopic investigations. There are, however, notable examples where the long-range behavior persists like in conventional dipolar ferromagnets [1, 2] and exotic spin-ice materials [3, 4]. In quantum optical platforms, long-range interactions are commonly present and there has been tremendous experimental advancements over the past decades. Indeed, among others, trapped ion systems [5–16] and neutral atoms in optical lattices [17–27] have gained vast attention as these platforms can realize one- and two-dimensional lattices with adaptable geometries and a mesoscopic number of entities offering high-fidelity control and read-out. This makes them viable candidates for versatile quantum simulators and scalable quantum computers [28–30]. Both platforms realize effective spin interactions which decay algebraically with distance. In neutral-atom platforms the decay exponent is fixed while it can be continuously tuned in trapped-ion systems. Recent progress ranges from the determination of molecular ground-state energies [15] and the realization of equilibrium [5, 25] and dynamical quantum phase transitions [12–14] to the direct observation of a topologically-ordered quantum spin liquid [26] and symmetry-protected topological phases realized on ladder geometries [22, 27].

The majority of numerical studies has focused on one-dimensional spin chains [31–46, 46–53] as well as two-dimensional systems directly related to Rydberg atom platforms with quickly decaying ($\sim r^{-6}$) long-range interactions [54–56]. One prominent exception is the long-range transverse-field Ising model (LRTFIM), which was recently analyzed on the two-dimensional square and triangular lattice with tunable long-range interactions [57–59]. Geometrically unfrustrated LRTFIMs in one and two dimensions are known from field-theoretical considerations to display three distinct regimes of quantum criticality between the high-field polarized phase and the low-field $\mathbb{Z}_2$-symmetry broken ground state: For short-range interactions the system exhibits nearest-neighbor criticality, for strong long-range interactions long-range mean-field behavior, and in-between continuously varying critical exponents [60–65].

Less is known about the quantum-critical behavior of systems with long-range interactions possessing a continuous symmetry. The antiferromagnetic spin-1/2 Heisenberg model is the most prominent example here where, however, only the one-dimensional chain has been investigated microscopically [36, 41, 42, 45, 47, 66]. For the short-range Heisenberg chain, the spontaneous breaking of its continuous $SU(2)$-symmetry is forbidden by the Hohenberg-Mermin-Wagner (HMW) theorem for finite temperature [67–70] and for zero temperature [71]. Here, one finds quasi long-range order with gapless fractional spinon excitations. The HMW theorem can be circumvented when unfrustrated long-range interactions are sufficiently strong, giving rise to a quantum phase transition to a Néel state with broken $SU(2)$-symmetry [36, 41, 42, 45, 47, 49, 66]. Interestingly, beyond the chain geometry, a recent work [72] has

studied an antiferromagnetic quasi one-dimensional two-leg quantum spin ladder with unfrustrated long-range Heisenberg interactions. Here, an exotic deconfined quantum critical point between the gapped short-range isotropic ladder with a non-local string order parameter and the Néel state with broken $SU(2)$-symmetry has been suggested [72]. The proposed transition goes therefore even beyond the established scenario of deconfined quantum criticality between two ordered phases with local order parameters [49, 73–76].

In this paper, we investigate two types of long-range quantum spin ladders with arbitrary ratios $\lambda$ of nearest-neighbor leg and rung exchange coupling and for arbitrary decay exponent $1 + \sigma$ of the long-range Heisenberg interaction. To this end, we extend the pCUT approach developed in Ref. [58] to generic observables and locate the critical breakdown of the rung-singlet phase in the $\sigma - \lambda$ parameter plane. This allows us to observe long-range mean-field behavior as well as a non-trivial regime of continuously varying critical exponents. We stress that the model studied in Ref. [72] is contained as one specific parameter line $\lambda = 1$ in our two-dimensional quantum phase diagram. From our findings and physical arguments we conclude that the investigated long-range Heisenberg quantum spin ladders do not show deconfined criticality.

## 2 Model: Quantum spin ladders with long-range interactions

We consider the spin-1/2 Hamiltonian

$$\mathcal{H} = J_\perp \sum_i \vec{S}_{i,1} \vec{S}_{i,2} - \sum_{i,\delta>0} \sum_{n=1}^{2} J_\parallel(\delta) \vec{S}_{i,n} \vec{S}_{i+\delta,n} - \sum_{i,\delta>0} J_\times(\delta) \left( \vec{S}_{i,1} \vec{S}_{i+\delta,2} + \vec{S}_{i,2} \vec{S}_{i+\delta,1} \right), \quad (1)$$

where the indices $i$ and $i + \delta$ denote the rung and the second index $n \in \{1, 2\}$ the leg of the ladder. The exchange parameters $J_\perp > 0$,

$$J_\parallel(\delta) = J_\parallel \frac{(-1)^\delta}{|\delta|^{1+\sigma}}, \qquad J_\times(\delta) = J_\times \frac{(-1)^{1+\delta}}{|1+\delta|^{1+\sigma}}, \quad (2)$$

couple spin operators on the rungs, legs, and diagonals, respectively. The distance-dependent coupling parameters $J_\parallel(\delta)$ and $J_\times(\delta)$ realize unfrustrated algebraically decaying long-range interactions which induce antiferromagnetic Néel order for sufficiently small $\sigma$. This decay exponent $\sigma$ can be tuned between the limiting cases of all-to-all interactions at $\sigma = -1$ and nearest-rung couplings at $\sigma = \infty$. Here, we focus on $\sigma \geq 0$ so that the energy of the system is extensive in the thermodynamic limit. We restrict to the limiting cases $\mathcal{H}_\parallel \equiv \mathcal{H}|_{J_\times=0}$ and $\mathcal{H}_\bowtie \equiv \mathcal{H}|_{J_\times=J_\parallel}$ illustrated in Fig. 1. In the following, we set $J_\perp = 1$ and introduce the perturbation parameter $\lambda \equiv J_\parallel$. Note, the Hamiltonian in Ref. [72] corresponds to $\mathcal{H}_\bowtie$ at $\lambda = 1$. In the limit of isolated rung dimers $\lambda = 0$, the ground state is given exactly by the product state of rung singlets

$$|s\rangle = \frac{1}{\sqrt{2}} \left( |\uparrow\downarrow\rangle - |\downarrow\uparrow\rangle \right), \quad (3)$$

and with localized rung triplets

$$|t_x\rangle = -\frac{1}{\sqrt{2}}(|\uparrow\uparrow\rangle - |\downarrow\downarrow\rangle), \quad |t_y\rangle = \frac{i}{\sqrt{2}}(|\uparrow\uparrow\rangle + |\downarrow\downarrow\rangle), \quad |t_z\rangle = \frac{1}{\sqrt{2}}(|\uparrow\downarrow\rangle + |\downarrow\uparrow\rangle) \quad (4)$$

as elementary excitations. For small $\lambda$ the ground state is adiabatically connected to this product state and the system is in the rung-singlet phase. The associated elementary excitations of the rung-singlet phase are gapped triplons [77] corresponding to dressed rung-triplet excitations. For $\sigma = \infty$ this holds for both spin ladders for any finite $\lambda$ and for $\mathcal{H}_\parallel$ at $\lambda = \infty$

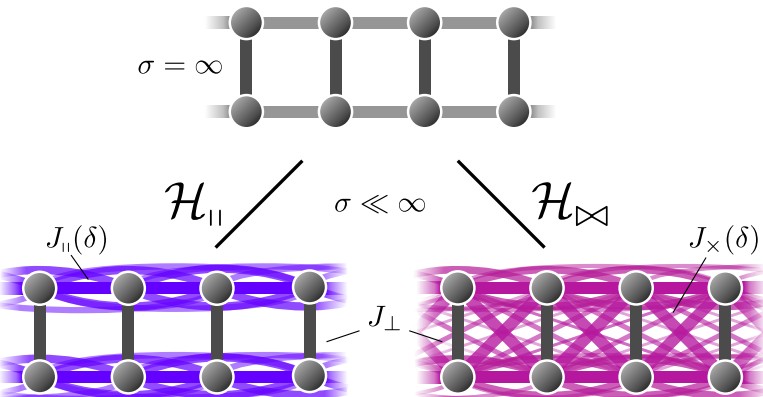

Figure 1: Illustration of the two quantum spin ladders with Heisenberg interaction on rung dimers ($\sim J_\perp$), between rung dimers along the legs ($\sim J_{\parallel}$) and along the diagonals ($\sim J_\times$). In the first row the common nearest-neighbor limit ($\sigma = \infty$) of both ladder models is shown while in the second row the two distinct spin ladders $\mathcal{H}_{\parallel}$ (left) and $\mathcal{H}_{\bowtie}$ (right) with long-range interactions $\sigma \ll \infty$ are sketched.

the system decouples into two spin-1/2 Heisenberg chains with gapless spinon excitations and a quasi long-range ordered ground state. The ground states at any finite $\lambda$ break a hidden $\mathbb{Z}_2 \times \mathbb{Z}_2$ symmetry and can be characterized by a non-local string order parameter [78–82].

Previous studies of the spin-1/2 Heisenberg chain [41, 42, 45, 49] and the two-leg ladder $\mathcal{H}_{\bowtie}$ for $\lambda = 1$ [72] with unfrustrated long-range interactions deduced a quantum phase transition towards Néel order with broken $SU(2)$-symmetry and thus circumventing the HMW theorem [67–71]. Further, Goldstone's theorem states that the spontaneous breaking of a continuous symmetry gives rise to massless Nambu-Goldstone modes [83–85], however, the same restriction applies and the theorem loses its validity in the presence of long-range interactions. Indeed, in the extreme case of an all-to-all coupling the ground-state energy becomes superextensive and the elementary excitations are gapped via a generalization of the Higgs mechanism [86].

## 3  Approach: High-order series expansions with pCUT

Our aim is to investigate the quantum critical breakdown of the rung-singlet phase. To this end, we extend the pCUT method [87, 88] to long-range Heisenberg interactions and determine high-order series expansions of relevant energies and observables in the thermodynamic limit about the limit of isolated rungs. It is then convenient to consider rung dimers as supersites and to reformulate the Hamiltonian (1) in terms of hard-core bosonic triplet creation and annihilation operators on rung dimers.

The pCUT method transforms the original Hamiltonian $\mathcal{H}$, perturbatively order by order in $\lambda$, into an effective Hamiltonian $\mathcal{H}_{\text{eff}}$ conserving the number of quasiparticles (QPs) which correspond to spin-one triplon excitations [77] – dressed rung triplets – in the rung-singlet phase. The same transformation has to be applied to observables, however, the quasiparticle-conserving property is lost. We can exploit the linked-cluster property [89] and perform the numerical calculations on finite topologically distinct graphs. In the end, the contributions on the finite graphs must be embedded on an infinite system to obtain the bulk properties which is equivalent to evaluating high-dimensional infinite sums that can be efficiently done by Monte Carlo integration [58].

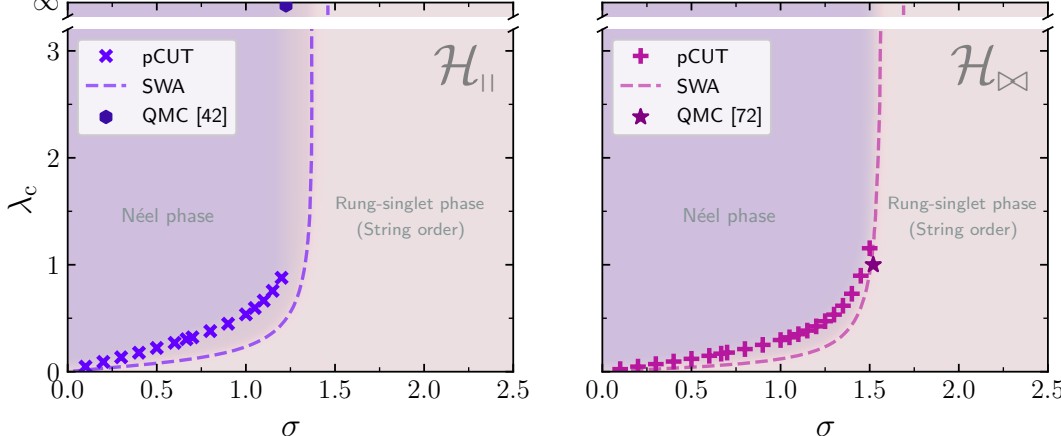

Figure 2: Quantum phase diagrams depicting the critical point $\lambda_c$ as a function of the decay exponent $\sigma$ for $\mathcal{H}_{\parallel}$ (left) and $\mathcal{H}_{\bowtie}$ (right). Crosses are determined by DlogPadé extrapolations of the one-triplon gap series from the pCUT method while dashed lines are extracted from the self-consistency condition for the staggered magnetization within linear spin-wave approximation (SWA). Comparing the left with the right plot, we observe that the Néel ordered phase sets in at smaller $\lambda$ or larger $\sigma$ exponents extending the Néel regime of $\mathcal{H}_{\bowtie}$. The hexagon point at $\lambda = \infty$ for $\mathcal{H}_{\parallel}$ corresponding to decoupled Heisenberg chains from Ref. [42] as well the star-shaped point along the $\lambda = 1$ line for $\mathcal{H}_{\bowtie}$ from Ref. [72] are consistent with our results.

Here, we investigate the zero- and one-triplon properties. The 0QP block of the effective Hamiltonian corresponds to the ground-state energy $\bar{E}_0$ while the 1QP block allows the calculation of the one-triplon gap $\Delta$ located at the critical momentum $k_c = \pi$. Further, we extended the pCUT approach for long-range interactions [58] to generic observables and determined the one-triplon spectral weight $\mathcal{S}^{1QP}(k_c)$. The latter corresponds to the one-triplon part of the Fourier transformed effective observable after the unitary transformation of the antisymmetric observable

$$\mathcal{O}_{i,z} = \frac{1}{2}(S^z_{i,1} - S^z_{i,2}) \tag{5}$$

on a rung dimer. We calculated high-order series of the control-parameter susceptibility $\chi \equiv -\frac{\mathrm{d}^2 \bar{E}_0}{\mathrm{d}\lambda^2}$ up to order 10 (6), the one-triplon gap $\Delta$ up to order 10 (7), and the one-triplon spectral weight $\mathcal{S}^{1QP}(k_c)$ up to order 9 (7) in $\lambda$ for $\mathcal{H}_{\parallel}$ ($\mathcal{H}_{\bowtie}$). See Appendix A for details on the pCUT approach.

The introduced quantities allow the extraction of critical exponents via the dominant power-law behavior

$$\chi \sim |\lambda - \lambda_c|^{-\alpha}, \tag{6}$$

$$\Delta \sim |\lambda - \lambda_c|^{z\nu}, \tag{7}$$

$$\mathcal{S}^{1QP}(k_c) \sim |\lambda - \lambda_c|^{-(2-z-\eta)\nu}, \tag{8}$$

close to the critical point $\lambda_c$ when the rung-singlet phase breaks down. The critical point and associated critical exponents can be directly determined from physical poles and associated residuals using (biased) DlogPadé extrapolants. The associated error bars should strictly be understood as the standard deviation from several extrapolants rather than rigorous errors. More detailed information on extrapolations can be found in Appendix B.

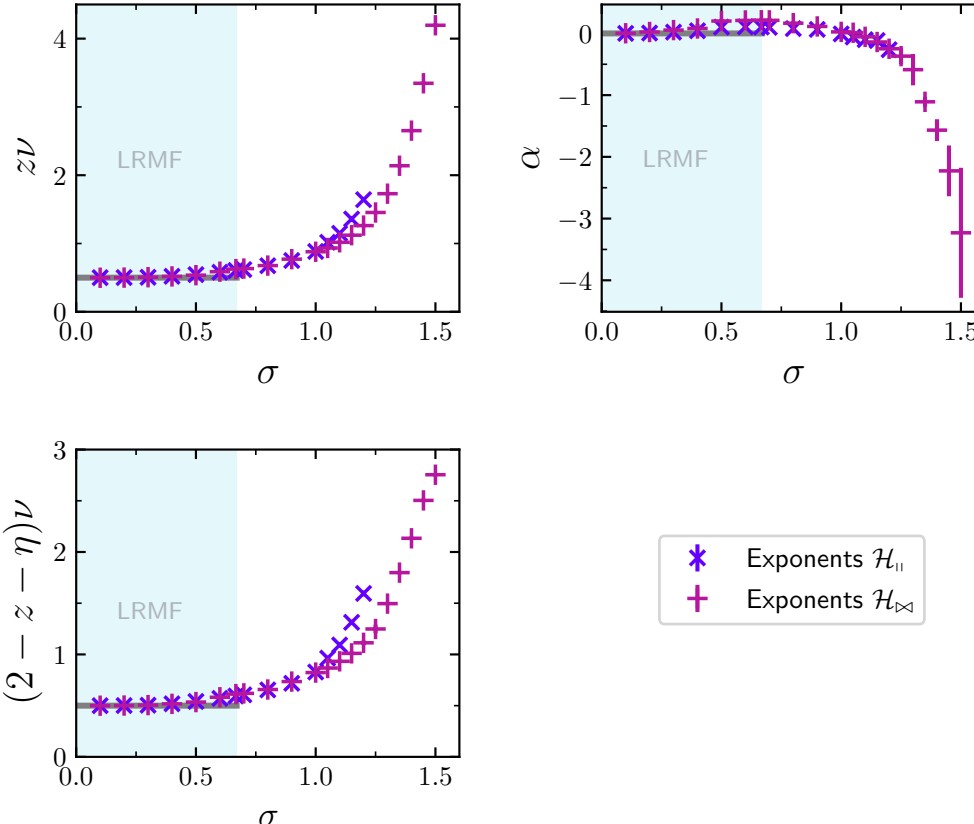

Figure 3: Critical exponents from Eqs. (6)-(8) determined by the pCUT approach as a function of the decay exponent $\sigma$ for both ladder models $\mathcal{H}_{\shortparallel}$ and $\mathcal{H}_{\bowtie}$. For $\sigma \leq 2/3$ the exponents coincide with the expected long-range mean-field values (shaded region). For $\sigma > 2/3$ they become continuously larger and start to diverge. While the critical exponent for both models match well for $\sigma \lesssim 2.1$, they start to deviate from each other for larger values of $\sigma$ but this can probably be attributed to the difference in $\sigma_*$.

## 4 Discussion of results

### 4.1 Quantum phase diagram

We determine the phase transition point $\lambda_c$ as a function of the decay exponent $\sigma$ by the quantum-critical breakdown of the rung-singlet phase and the accompanied closing of the one-triplon gap. The corresponding quantum phase diagram is shown in Fig. 2 for $\mathcal{H}_{\shortparallel}$ and $\mathcal{H}_{\bowtie}$. In accordance with the HMW theorem, a quantum phase transition can be ruled out from one-loop renormalization group (RG) for $\sigma > 2$ [60], since the one-dimensional $O(3)$ quantum rotor model can be mapped to the low-energy physics of the dimerized antiferromagnetic Heisenberg ladder [90]. At small $\sigma \lesssim 0.7$ ($\sigma \lesssim 1.0$) for $\mathcal{H}_{\shortparallel}$ ($\mathcal{H}_{\bowtie}$) the critical point $\lambda_c$ shifts linearly towards larger $\lambda$ with increasing $\sigma$. The gap closes earlier for $\mathcal{H}_{\bowtie}$ in agreement with expectations since the additional diagonal interactions further stabilize the antiferromagnetic Néel order. For larger $\sigma$ the critical points start to deviate from the linear behavior and bend upwards towards larger critical points until eventually DlogPadé extrapolations break down when the critical point shifts away significantly from the radius of convergence of the series.

We complement the pCUT approach with linear spin-wave calculations similar to the ones in Refs. [41, 42]. Exploiting the fact that spin-wave theory is expected to work only in the

Néel ordered phase, we can determine the quantum-critical line from a consistency condition for the staggered magnetization (see also Appendix C). Linear spin-wave theory allows us to qualitatively determine the extent of the Néel ordered phase in the whole parameter regime. Indeed, we find for small $\lambda$ that linear spin-wave theory agrees well with the pCUT findings and we also observe that the Néel regime extends to smaller $\lambda$ and larger $\sigma$ for $\mathcal{H}_{\bowtie}$ due to the additional diagonal interactions. In the limit $\lambda = \infty$ where the pCUT series expansion does not provide any meaningful results we locate an upper critical bound $\sigma_*$ inline with the absence of criticality at large enough $\sigma$. This upper bound corresponds therefore to the lower critical dimension. In fact, for $\mathcal{H}_{\parallel}$ at $\lambda = \infty$ in the limit of decoupled Heisenberg chains we recover the spin-wave dispersion in Ref. [42] yielding $\sigma_*^{\mathrm{SW}} \approx 1.46$ and for $\mathcal{H}_{\bowtie}$ at $\lambda = \infty$ we find $\sigma_*^{\mathrm{SW}} \approx 1.69$.

Moreover, all our data is consistent with $\sigma_* = 1.225(25)$ at $\lambda = \infty$ from Ref. [42] for $\mathcal{H}_{\parallel}$ and with $\sigma_c \approx 1.52$ at $\lambda_c = 1$ for $\mathcal{H}_{\bowtie}$ in Ref. [72] as depicted in Fig. 2. Besides this, the critical exponents in the long-range mean-field realm discussed below are in very good agreement with field-theoretical expectations. However, the distinct values for $\sigma_*$ from spin-wave calculations and QMC [42] consistent with the pCUT results for both ladder models are unexpectedly at significant smaller values than predicted from the one-dimensional long-range $O(3)$ quantum rotor model with $\sigma_* = 2$ [61, 62, 91, 92].

## 4.2  Critical exponents

We extract the critical exponents according to Eqs. (6)-(8) from DlogPadé extrapolants of the perturbative series. The exponents are depicted in Fig. 3 as a function of the decay exponent $\sigma$. The long-range mean-field regime (LRMF) is expected to extend to $\sigma_{\mathrm{uc}} = 2/3$ [60]. The extracted exponents agree well with expected long-range mean-field exponents, although the presence of multiplicative logarithmic corrections to the dominant power-law behavior at the upper critical dimension $d_{\mathrm{uc}} = 3\sigma/2$ negatively affects the accuracy of the deduced critical exponents around $\sigma = 2/3$ as known from the LRTFIM [38, 58]. Estimates for multiplicative logarithmic critical exponents can be found in Appendix B. Excluding the $\alpha$-exponent the critical exponents deviate less than $1.1\,\%$ ($1.3\,\%$) deep in the long-range regime $\sigma \leq 0.3$ for $\mathcal{H}_{\parallel}$ ($\mathcal{H}_{\bowtie}$). For $\sigma > 2/3$ we observe continuously varying exponents which seem to diverge for $\sigma \to \sigma_*$. In terms of the gap closing this can be understood from the nearest-neighbor limit where the gap does not close but with the increasingly stronger long-range interactions the finite gap is lowered until eventually the gap closes. Further strengthening the long-range interactions shifts the critical point from infinity to smaller values and thus continuously tuning the exponent $z\nu$ from infinity to smaller values as the gap closes increasingly steep. In the region $\sigma \gtrsim 1.1$ for $\mathcal{H}_{\parallel}$ ($\sigma \gtrsim 1.2$ for $\mathcal{H}_{\bowtie}$) close to $\sigma_*$ it becomes difficult to extrapolate the gap series as the critical point starts to shift quickly towards $\lambda = \infty$. This negatively affects the accuracy of the exponent estimates.

Using the three critical exponents shown Fig. 3, one can apply the scaling relations

$$
\begin{aligned}
\gamma &= (2 - \eta)\nu, \\
\gamma &= \beta(\delta - 1), \\
2 &= \alpha + 2\beta + \gamma,
\end{aligned} \tag{9}
$$

as well as the hyperscaling relation

$$
2 - \alpha = \left( \frac{d}{\varrho} + z \right)\nu, \tag{10}
$$

with the pseudocritical exponent $\varrho$. The hyperscaling relation was only recently generalized to be valid above the upper critical dimension [40]. This allows us to directly derive all canonical



Figure 4: Canonical critical exponents obtained from (hyper-) scaling relations as a function of the decay exponent $\sigma$. The critical exponents are in good agreement with expectations in the long-range mean-field regime (shaded region) and show continuously varying exponents for $\sigma > 2/3$. While some critical exponents appear to diverge others seem to go to a constant value for increasing $\sigma$. For some exponents the error bars become larger for $\sigma \approx \sigma_*$.

critical exponents for any $\sigma$ (see Appendix D). The canonical critical exponents are depicted in Fig. 4 for $\mathcal{H}_{\parallel}$ and $\mathcal{H}_{\bowtie}$. In the long-range mean-field regime the exponents agree well with the expectations. The exponents $\beta$ and $1/\delta$ around the upper critical dimension show larger deviations which we attribute to error propagation due to the presence of multiplicative loga-

rithmic corrections. While the critical exponent $\gamma$ diverges for larger values of $\sigma$, the critical exponent $\nu$ approaches a constant value $\nu \approx 1$. The exponent $1/\delta$ goes to $-0.125$ in this limit and we attribute this to a systematic error arising from the diverging critical exponents close to $\sigma_*$. Instead, the correct physical limit might be 0 since a sign change of $1/\delta$ is unphysical. For the exponents $\beta$, $z$, and $\eta$ the uncertainty in the regime $\sigma \gtrsim 1.2$ becomes large due to error propagation and it is hard to make precise statements in the vicinity of $\sigma_*$. Nonetheless, we find that $\eta$ differs from the linear behavior $\eta = 2 - \sigma$ expected by field theory for $\sigma < \sigma_*$ [61, 91, 92] going faster to zero (until unphysically negative values are obtained) but in agreement with our previous finding that $\sigma_*$ is smaller than expected by the long-range $O(3)$ quantum rotor model. Interestingly, also Heisenberg chains with long-range interactions differ from the field-theoretical expectation $\eta = 2 - \sigma$. However, in Ref. [42] they observe $z < 1$ and $\eta \geq 2 - \sigma$ while we find $z > 1$ and $\eta \leq 2 - \sigma$.

Comparing the above results with Ref. [72] for $\mathcal{H}_{\bowtie}$ at $\lambda = 1$ we find that the exponent $\nu = 1.8$ at about $\sigma_c \approx 1.5$ is inconsistent with our result $\nu = 0.97(7)$ for all $\sigma > 1.0$ which appears to be particularly well converged compared to other critical exponents. Furthermore, the monotonously increasing exponent $z > 1$ for $\sigma > 1.1$ is not in line with a proposed deconfined critical point with $z = 1$ at $\sigma \approx 1.5$. Our finding of continuously varying exponents reminiscent of the criticality of the unfrustrated LRTFIM [38, 58, 60–65] raises the question why this specific point should display deconfined criticality, particularly considering that despite the presence of a non-local string order parameter the rung singlet-phase of both models $\mathcal{H}_{\parallel}$ and $\mathcal{H}_{\bowtie}$ for all relevant $\lambda$ is not topologically protected but trivially connected to the product state of rung singlets [93].

## 5 Conclusions

We investigated the quantum-critical behavior of two unfrustrated two-leg quantum spin ladders with long-range Heisenberg interactions by applying and extending the pCUT method in combination with classical Monte Carlo integration that allows us to determine relevant energies and observables in the thermodynamic limit. From the closing of the one-triplon gap we determined the phase diagram in the $\sigma - \lambda$ plane for both spin ladders. Interestingly, we find lower critical dimensions $\sigma_* < 2$ unlike $\sigma_* = 2$ from field-theoretical predictions for the one-dimensional long-range $O(3)$ quantum rotor model, but in agreement with known results [42] from the isolated chain limit. By generalizing the pCUT approach for long-range systems to generic observables, we calculated the ground-state energy and the one-triplon spectral weight so that we were able to extract the full set of critical exponents as a function of the decay exponent using appropriate extrapolation techniques. A non-trivial regime of continuously varying critical exponents as well as long-range mean-field behavior was observed. From these findings and the fact that the rung-singlet phase is not topologically protected we conclude the absence of deconfined criticality in the investigated models. However, quantum phase transitions between phases with local order and non-local string order parameters, where the latter phase is indeed topologically protected, should be investigated in the future as such systems might realize exotic properties like deconfined criticality. The spin-one Heisenberg chain with unfrustrated long-range interactions should therefore be very interesting to look at. Our approach can further be naturally extended to gapped phases of higher-dimensional Heisenberg systems with long-range interactions, e.g., bilayer geometries. This opens a completely unexplored playground for future research.

# Acknowledgements

P.A. and K.P.S. thank M. Mühlhauser for providing the graph files, J. A. Koziol for fruitful discussions and thankfully acknowledge the scientific support and HPC resources provided by the Erlangen National High Performance Computing Center (NHR@FAU) of the Friedrich-Alexander-Universität Erlangen-Nürnberg (FAU).

**Funding information** We gratefully acknowledge the support by the Deutsche Forschungsgemeinschaft (DFG, German Research Foundation) – Project-ID 429529648—TRR 306 QuCoLiMa ("Quantum Cooperativity of Light and Matter") as well as the Munich Quantum Valley, which is supported by the Bavarian state government with funds from the Hightech Agenda Bayern Plus. The hardware of NHR@FAU is funded by the German Research Foundation DFG.

**Author contributions** P.A. performed the spin-wave calculations and the numerical simulations. P.A. analyzed the results with assistance from K.P.S. who supervised the project. Both authors contributed equally to the writing of the manuscript.

**Data availability** The raw data is available on Zenodo at: https://doi.org/10.5281/zenodo.7918837. The code used to generate the numerical results presented in this paper can be made available by Patrick Adelhardt (patrick.adelhardt@fau.de) upon reasonable request.

**Competing interests** The authors declare no competing interests.

# A High-order series expansion

In the following, we provide a description of the high-order series expansions approach using the pCUT method along the same lines as in previous studies on the LRTFIM [39, 40, 58, 94]. The approach can be generalized to observables which allows us to determine the entire set of critical exponents.

## A.1 The pCUT method

To apply the pCUT method [87, 88] it must be possible to describe the problem under consideration with a Hamiltonian of the form

$$\mathcal{H} = \mathcal{H}_0 + \mathcal{V} = E_0 + \mathcal{Q} + \sum_{\delta > 0}^{\infty} \lambda(\delta) \mathcal{V}(\delta), \tag{11}$$

with an unperturbed Hamiltonian $\mathcal{H}_0$ with equidistant spectrum that is bounded from below and a perturbation $\mathcal{V}$. We bring the spin-ladder Hamiltonian into this form by interpreting the Hamiltonian as a system of coupled supersites (dimers) and introducing hardcore bosonic triplet (creation) annihilation operators $t_{i,\rho}^{(\dagger)}$ (creating) annihilating local triplets with flavor $\rho \in \{x, y, z\}$ on rung $i$ [95, 96]. The unperturbed part becomes $\mathcal{H}_0 = E_0 + \mathcal{Q}$ with $E_0 = -3/4 N_{\text{rung}}$ the unperturbed ground-state energy, $N_{\text{rung}}$ the number of rungs, and $\mathcal{Q} = \sum_{i,\rho} t_{i,\rho}^{\dagger} t_{i,\rho}$ counting the number of triplet quasiparticles (QPs). For long-range systems the perturbation $\mathcal{V}$ can be written as a sum between interacting processes of distance $\delta$ with a

distance-dependent expansion parameter $\lambda(\delta)$. Also, the perturbation must decompose into

$$\mathcal{V} = \sum_{m=-N}^{N} T_m = \sum_{m=-N}^{N} \sum_l \tau_{m,l}, \tag{12}$$

where the operators $T_m$ change the system's energy by $m$ energy quanta such that $[\mathcal{Q}, T_m] = m T_m$. For the spin ladder Hamiltonian we have $m \in \{0, \pm 2\}$. The operator $T_m$ decomposes into a sum of local operators $\tau_{m,l}$ on a link $l$ connecting different sites of the underlying lattice. When the above prerequisites are fulfilled the pCUT method unitarily transforms the original Hamiltonian, order by order in the perturbation parameter $\lambda$, to an effective, quasiparticle-conserving Hamiltonian $\mathcal{H}_{\text{eff}}$ reducing the complicated many-body problem to an easier effective few-body problem. The effective Hamiltonian in a generic form for an arbitrary number of expansion parameters $\lambda_i$ is then given by

$$\mathcal{H}_{\text{eff}} = \mathcal{H}_0 + \sum_{\sum_j^{N_\lambda} n_j = k}^{\infty} \lambda_1^{n_1} \dots \lambda_{N_\lambda}^{n_{N_\lambda}} \sum_{\substack{\dim(\boldsymbol{m})=k, \\ \sum_i m_i = 0}} C(\boldsymbol{m}) \, T_{m_1} \dots T_{m_k}, \tag{13}$$

where the coefficients $C(\boldsymbol{m})$ are exactly given by rational numbers and the condition $\sum_i m_i = 0$ enforces the quasiparticle conservation $[\mathcal{Q}, \mathcal{H}_{\text{eff}}] = 0$. Analogously, an effective observable is given by

$$\mathcal{O}_{\text{eff}} = \sum_{\sum_j^{N_\lambda} n_j = k}^{\infty} \lambda_1^{n_1} \dots \lambda_{N_\lambda}^{n_{N_\lambda}} \sum_{i=1}^{k+1} \sum_{\dim(\boldsymbol{m})=k} \tilde{C}(\boldsymbol{m}; i) \, T_{m_1} \dots T_{m_{i-1}} \mathcal{O} T_{m_i} \dots T_{m_k}, \tag{14}$$

with the rational coefficient $\tilde{C}(\boldsymbol{m}; i)$. In contrast to the effective Hamiltonian the effective observable is not quasiparticle conserving. The effective Hamiltonian and observables are generally independent of the exact form of the original Hamiltonian as long as the pCUT prerequisites are satisfied. To bring $\mathcal{H}_{\text{eff}}$ and $\mathcal{O}_{\text{eff}}$ into normal-ordered form, a model-dependent extraction process must be applied. For long-range interactions this is done most efficiently by a full-graph decomposition.

## A.2   Graph decomposition

We apply the effective quantities to finite, topologically distinct graphs to bring them into normal-ordered structure. We refer to this approach as a linked-cluster expansion implemented as a full-graph decomposition. The underlying principle is the linked-cluster theorem which states that only linked processes have an overall contributions to cluster-additive quantities [89]. Since the effective pCUT Hamiltonian and observables are cluster-additive quantities we can reformulate Eqs. (13) and (14) as

$$\mathcal{H}_{\text{eff}} = \mathcal{H}_0 + \sum_{\sum_j^{N_\lambda} n_j = k}^{\infty} \lambda_1^{n_1} \dots \lambda_{N_\lambda}^{n_{N_\lambda}} \sum_{\substack{\dim(\boldsymbol{m})=k, \\ \sum_i m_i = 0}} \sum_{\substack{\mathcal{G}, \\ |\mathcal{E}_\mathcal{G}| \le k}} C(\boldsymbol{m}) \sum_{\substack{l_1, \dots, l_k, \\ \bigcup_{i=1}^k l_i = \mathcal{G}}} \tau_{m_1, l_1} \dots \tau_{m_k, l_k}, \tag{15}$$

$$\mathcal{O}_{\text{eff}} = \sum_{\sum_j^{N_\lambda} n_j = k}^{\infty} \lambda_1^{n_1} \dots \lambda_{N_\lambda}^{n_{N_\lambda}} \sum_{i=1}^{k+1} \sum_{\dim(\boldsymbol{m})=k} \sum_{\substack{\mathcal{G}, \\ |\mathcal{E}_\mathcal{G}| \le k}} \tilde{C}(\boldsymbol{m}; i)$$
$$\times \sum_{\substack{l_1, \dots, l_k, \\ \bigcup_{i=1}^k l_i \cup x = \mathcal{G}}} \tau_{m_1, l_1} \dots \tau_{m_{i-1}, l_{i-1}} \mathcal{O}_x \tau_{m_i, l_i} \dots \tau_{m_k, l_k}, \tag{16}$$

where the sum over $\mathcal{G}$ runs over all possible simple connected graphs of perturbative order $k \geq |\mathcal{E}_\mathcal{G}|$. A graph $\mathcal{G}$ is a tuple $(\mathcal{E}_\mathcal{G}, \mathcal{V}_\mathcal{G})$ consisting of an edge or link set $\mathcal{E}_\mathcal{G}$ with $|\mathcal{E}_\mathcal{G}|$ edges and a set of vertices or sites $\mathcal{V}_\mathcal{G}$ with $|\mathcal{V}_\mathcal{G}|$ vertices. The conditions $\bigcup_{i=1}^{k} l_i = \mathcal{G}$ and $\bigcup_{i=1}^{k} l_i \cup x = \mathcal{G}$ arising from the linked-cluster theorem ensure that the cluster made up of active links and sites during a process must match with the edge and vertex set of a simple connected graph $\mathcal{G}$. Note, we generalized the notation for observables $\mathcal{O}_x$ where the index $x$ can either refer to a site (local observable) or a link (non-local observable). Thus, we can set up a full-graph decomposition applying the effective quantities to a set of finite, topologically distinct, simple connected graphs.

In the standard approach one would identify different expansion parameters with link colors which serve as another topological attribute in the classification of graphs. However, this approach fails for long-range interactions because every coupling parameter $\lambda(\delta)$ between sites of distance $\delta$ would be associated to a distinct link color and the number of graphs would already be infinite in first order of perturbation. We can overcome this obstacle by introducing white graphs [89] where different link colors are ignored in the topological classification of graphs and instead additional information is tracked during the calculation on white graphs. In particular, every link on a graph is associated with a distinct expansion parameter $\lambda_n^\mathcal{G}$ yielding a multivariable polynomial after applying the effective quantities to the graph. Only during the embedding on the lattice the proper link color is reintroduced by replacing the expansion parameters of the polynomial by the actual coupling strength for each realization decaying algebraically with the distance between interacting sites.

### A.3 Monte Carlo embedding

Since we describe the ladder system in the language of rung dimers as super sites the graph contributions from the linked-cluster expansion must be embedded into a one-dimensional chain to determine the values of physical quantities $\kappa = \sum_m c_m^{(\kappa)} \lambda^m$ as a high-order series in the thermodynamic limit. Due to the infinite range of the algebraically decaying interactions every graph can be embedded infinitely many times at any order of perturbation. For each realization of a graph on the infinite chain the generic couplings $\lambda_n^\mathcal{G}$ in the multivariable polynomial corresponding to distinct edges is substituted by the true coupling strength $\lambda(-1)^\delta |\delta|^{-1-\sigma}$ or $\lambda(-1)^{1+\delta}|1+\delta|^{-1-\sigma}$ between graph vertices on sites $i$ and $i+\delta$ on the chain. For a prefactor $c_m$ in the high-order series only (reduced) contributions from graphs with up to $m$ links and $m+1$ sites can contribute. See Ref. [89] for remarks about reduced quantities. We can write explicitly

$$c_m^{(\kappa)} = \sum_{N=2}^{m+1} \sum_a f_N(a) = \sum_{N=2}^{m+1} S[f_N], \tag{17}$$

where the first sum goes over the number of vertices and the second sum over all possible configurations excluding embeddings with overlapping vertices. The integrand $f_N$ combines all contributions from graphs with the same number of vertices $N$ since the $m-1$ sums contained in the sum $\sum_a$ are identical for graphs with the same number of vertices. The integration of these high-dimensional infinite nested sums $S[\cdot]$ quickly becomes very challenging when the perturbative order increases. It is essential to use Monte Carlo (MC) integration to evaluate these sums since MC techniques are known to be well suited for high-dimensional problems. We take a Markov-chain Monte Carlo approach to sample the configuration space [58]. The fundamental moves consist of randomly selecting and moving graph vertices on the chain. For every embedding the integrands $f_N$ are evaluated with the correct couplings and added up to the overall contributions [58].

## A.4 Derivation of physical quantities

After having established the theoretical framework of the pCUT approach, we derive the physical quantities used in this paper. We start by stating the normal-ordered effective one-triplon (1QP) Hamiltonian given by

$$\mathcal{H}_{\text{eff}}^{1\text{QP}} = \bar{E}_0 + \sum_{\rho} \sum_{j,\delta \geq 0} a_\delta (t_{j,\rho}^\dagger t_{j+\delta,\rho} + \text{h.c.}), \tag{18}$$

with the ground-state energy $\bar{E}_0$ and the 1QP hopping amplitudes $a_\delta$. We determine the ground-state energy

$$\bar{E}_0 = \sum_m c_m^{(\bar{E}_0)} \lambda^m, \tag{19}$$

in the thermodynamic limit as a high-order series in the perturbation parameter $\lambda$ using the above described procedure where the general white-graph contributions must by embedded into the infinite chain of dimer supersites using Monte Carlo summation yielding estimates for $c_m^{(\bar{E}_0)}$. The control parameter susceptibility can be directly obtained using

$$\chi = -\frac{\mathrm{d}^2 \bar{E}_0}{\mathrm{d}\lambda^2}. \tag{20}$$

To get the one-triplon excitation gap as a high-order series, we remember that Eq. (18) can be diagonalized by transforming into momentum space, yielding

$$\tilde{\mathcal{H}}_{\text{eff}}^{1\text{QP}} = \bar{E}_0 + \sum_{k,\rho} \omega(k) t_{k,\rho}^\dagger t_{k,\rho} \quad \text{with} \quad \omega(k) = a_0 + 2\sum_{\delta > 0} a_\delta \cos(k\delta), \tag{21}$$

so the one-triplon gap is given by

$$\Delta = \min_k \omega(k) = \omega(k_c) = \sum_m c_m^{(\Delta)} \lambda^m, \tag{22}$$

with the critical momentum $k_c = \pi$ for antiferromagnetic interactions. Analogously to the ground-state energy, we determine Monte Carlo estimates for $c_m^{(\Delta)}$. Last, we introduce the dynamic structure factor

$$\mathcal{S}_{\rho,\rho}(k,\omega) = \frac{1}{2\pi N} \sum_{i,j} \int_{-\infty}^{\infty} \mathrm{d}t \, \exp\{\mathrm{i}[\omega t - k(j-i)]\} \langle \mathcal{O}_{i,\rho}(t) \mathcal{O}_{j,\rho}(0) \rangle, \tag{23}$$

with the observable defined as the antisymmetric combination of spin operators

$$\mathcal{O}_{i,\rho} = \frac{1}{2}(S_{i,1}^\rho - S_{i,2}^\rho) = \frac{1}{2}(t_{i,\rho}^\dagger + t_{i,\rho}) \tag{24}$$

of flavor $\rho$ on a rung $i$. We now follow the steps in Ref. [97]. Integrating out the energy $\omega$, one can express the structure factor in the effective basis as a sum over spectral weights $\mathcal{S}_{\rho,\rho}^{n\text{QP}}$ with fixed quasi-particle number

$$\mathcal{S}_{\rho,\rho}(k) = \sum_n \mathcal{S}_{\rho,\rho}^{n\text{QP}}(k). \tag{25}$$

By changing into the Heisenberg picture we eventually arrive at

$$\mathcal{S}_{\rho,\rho}^{1\text{QP}}(k) = \left| \langle t_{k,\rho} | \mathcal{O}_{\text{eff},\rho}^{1\text{QP}}(k) | \text{ref} \rangle \right|^2 = |s(k)|^2, \tag{26}$$

for the one-triplon spectral weight, where $|\text{ref}\rangle = \bigotimes_i |s_i\rangle$ is the unperturbed rung-singlet ground state and $\left|t_{k,\rho}\right\rangle$ is the one-triplon state with momentum $k$ and flavor $\rho$. In second quantization the effective observable restricted to the one-triplon channel can be expressed as

$$\mathcal{O}^{1\text{QP}}_{\text{eff},\rho}(k) = s(k)(t^\dagger_{k,\rho} + t_{k,\rho}).\tag{27}$$

Due to the SU(2)-symmetry one has $\mathcal{S}_{x,x} = \mathcal{S}_{y,y} = \mathcal{S}_{z,z}$, so we restrict in the following to $\rho = z$ and calculate $\mathcal{S}^{1\text{QP}} \equiv \mathcal{S}^{1\text{QP}}_{z,z}$. When we fix $k = k_c$ we can obtain a high order series of

$$s(k_c) = \sum_m c_m^{(s(k_c))} \lambda^m,\tag{28}$$

from the Monte Carlo estimates of $c_m^{(s(k_c))}$ and determine one-triplon spectral weight simply by calculating the absolute square.

# B   DlogPadé extrapolations

To extract the quantum-critical point including critical exponents from the pCUT method even beyond the radius of convergence of the pure high-order series we use DlogPadé extrapolations. For a detailed description on DlogPadés and its application to critical phenomena we refer to Refs. [98, 99]. The Padé extrapolant of a physical quantity $\kappa$ given as a perturbative series is defined as

$$P[L,M]_\kappa = \frac{P_L(\lambda)}{Q_M(\lambda)} = \frac{p_0 + p_1\lambda + \cdots + p_L\lambda^L}{1 + q_1\lambda + \cdots + q_M\lambda^M},\tag{29}$$

with $p_i, q_i \in \mathbb{R}$ and the degrees $L$, $M$ of $P_L(x)$ and $Q_M(x)$ with $r \equiv L + M$, i.e., the Taylor expansion of Eq. (29) about $\lambda = 0$ up to order $r$ must recover the quantity $\kappa$ up to the same order. For DlogPadé extrapolants we introduce

$$\mathcal{D}(\lambda) = \frac{\mathrm{d}}{\mathrm{d}\lambda} \ln(\kappa) \equiv P[L,M]_\mathcal{D},\tag{30}$$

the Padé extrapolant of the logarithmic derivative $\mathcal{D}$ with $r - 1 = L + M$. Thus the DlogPadé extrapolant of $\kappa$ is given by

$$\mathrm{d}P[L,M]_\kappa = \exp\left(\int_0^\lambda P[L,M]_\mathcal{D}\,\mathrm{d}\lambda'\right).\tag{31}$$

Given a dominant power-law behavior $\kappa \sim |\lambda - \lambda_c|^{-\theta}$, an estimate for the critical point $\lambda_c$ can be determined by excluding spurious extrapolants and analyzing the physical pole of $P[L,M]_\mathcal{D}$. If $\lambda_c$ is known, we can define biased DlogPadés by the Padé extrapolant

$$\theta^* = (\lambda_c - \lambda)\frac{\mathrm{d}}{\mathrm{d}\lambda} \ln(\kappa) \equiv P[L,M]_{\theta^*}.\tag{32}$$

In the unbiased as well as the biased case we can extract estimates for the critical exponent $\theta$ by calculating the residua

$$\begin{aligned} \theta_{\text{unbiased}} &= \operatorname{Res} P[L,M]_\mathcal{D}|_{\lambda=\lambda_c}, \\ \theta_{\text{biased}} &= \operatorname{Res} P[L,M]_{\theta^*}|_{\lambda=\lambda_c}. \end{aligned}\tag{33}$$

At the upper critical dimension $\sigma = 2/3$ multiplicative logarithmic corrections to the dominant power law behavior

$$\kappa \sim |\lambda - \lambda_c|^{-\theta} (\ln(\lambda - \lambda_c))^{p_\theta},\tag{34}$$

Table 1: Multiplicative logarithmic corrections $p_\theta$ at the upper critical dimension $\sigma_{\text{uc}} = 2/3$ associated to the ground-state energy $p_\alpha$, the 1QP excitation gap $p_{z\nu}$, and the 1QP spectral weight $p_{(2-z-\eta)\nu}$. Expected values from field-theoretical consideration are read of from Refs. [100, 101].

|  | Multiplicative correction | | |
|---|---|---|---|
|  | $p_\alpha$ | $p_{z\nu}$ | $p_{(2-z-\eta)\nu}$ |
| Field-theoretical predictions | $\frac{1}{11} \approx 0.091$ | $-\frac{5}{22} \approx -0.227$ | ? |
| $\mathcal{H}_{\parallel}$ | 0.453(6) | -0.309(13) | 3.94(11) |
| $\mathcal{H}_{\bowtie}$ | 0.533(16) | -0.374(19) | 3.77(12) |

in the vicinity of the quantum-critical point $\lambda_c$ are present. By biasing the critical point $\lambda_c$ and the exponent $\theta$ to its mean-field value, we define

$$p_\theta^* = -\ln(1 - \lambda/\lambda_c)[(\lambda_c - \lambda)\mathcal{D}(\lambda) - \theta] \equiv P[L, M]_{p_\theta^*}, \qquad (35)$$

such that we can determine an estimate for $p_\theta$ by again calculating the residuum of the Padé extrapolants $P[L, M]_{p_\theta^*}$. Note, for all quantities we calculate a large set of DlogPadé extrapolants with $L + M = r' \leq r$, exclude defective extrapolants, and arrange the remaining DlogPadés in families with $L - M = \text{const}$. Although individual extrapolations deviate from each other, the quality of the extrapolations increases with the order of perturbation as members of different families but mutual order $r'$ converge. To systematically analyze the quantum-critical regime, we take the mean of the highest order extrapolants of different families with more than one member. Here, we use DlogPadé extrapolation for the gap series to determine the critical point $\lambda_c$ and the critical exponent $z\nu$. We then apply biased DlogPadé extrapolation with $\lambda_c$ from the one-tripolon gap to obtain estimates for $\alpha$ and $2 - z - \eta$ via the series of the susceptibility and the one-triplon spectral weight.

Multiplicative logarithmic exponents to the power law scaling for both ladder models $\mathcal{H}_{\parallel}$ and $\mathcal{H}_{\bowtie}$ can be found in Table 1. We find estimates in the correct order of magnitude for $p_\alpha$ and $p_{z\nu}$ with better estimates for the logarithmic correction exponent of the gap. For $p_{(2-z-\eta)}$ there are no field-theoretical predictions directly available. Note, it is extremely challenging to accurately extract logarithmic corrections since the extracted values are very sensitive on the position of the critical point and DlogPadés are known to overestimate the critical value [39].

## C  Linear spin-wave calculations

We supplement the critical behavior determined by the pCUT approach with critical points from linear spin-wave approximation. As spin-wave theory considers fluctuations about the classical ground state it is certainly valid in the Néel-ordered phase of the long-range Heisenberg ladders. We start by mapping the spin operators to boson creation and annihilation operators using the Holstein-Primakoff transformation up to linear order in the boson operators. For the antiferromagnetic Heisenberg spin ladder the system must be divided into two sublattices constituting the expected antiferromagnetic Néel order for strong long-range interactions. The

transformation thus reads

$$
\begin{aligned}
S^z_{i,1} &= S - a^\dagger_{i,1} a_{i,1}, & S^-_{i,1} &\approx \sqrt{2S}\, a^\dagger_{i,1}, & S^+_{i,1} &\approx \sqrt{2S}\, a_{i,1}, \\
S^z_{i,2} &= b^\dagger_{i,2} b_{i,2} - S, & S^-_{i,2} &\approx \sqrt{2S}\, b_{i,2}, & S^+_{i,2} &\approx \sqrt{2S}\, b^\dagger_{i,2}, \\
S^z_{j,1} &= b^\dagger_{j,1} b_{j,1} - S, & S^-_{j,1} &\approx \sqrt{2S}\, b_{j,1}, & S^+_{j,1} &\approx \sqrt{2S}\, b^\dagger_{j,1}, \\
S^z_{j,2} &= S - a^\dagger_{j,2} a_{j,2}, & S^-_{j,2} &\approx \sqrt{2S}\, a^\dagger_{j,2}, & S^+_{j,2} &\approx \sqrt{2S}\, a_{j,2},
\end{aligned}
\tag{36}
$$

with $i$ odd and $j$ even rungs. Inserting these identities into the Hamiltonian $\mathcal{H}_\parallel$, neglecting quartic terms and Fourier transforming the problem, we arrive at

$$
\mathcal{H}^{\mathrm{SW}}_\parallel \approx \mathrm{const.} + S \sum_k \Bigg\{ \sum_\nu \Big[ (\gamma - f(k)) \big( a^\dagger_{k,\nu} a_{k,\nu} + b^\dagger_{-k,\nu} b_{-k,\nu} \big) + g(k) \big( a_{k,\nu} b_{-k,\nu} + a^\dagger_{k,\nu} b^\dagger_{-k,\nu} \big) \Big]
$$
$$
+ a_{k,1} b_{-k,2} + a_{k,2} b_{-k,1} + a^\dagger_{k,1} b^\dagger_{-k,2} + a^\dagger_{k,2} b^\dagger_{-k,1} \Bigg\}.
\tag{37}
$$

Incorporating the long-range couplings for an infinite chain into the prefactors we can define the quantities

$$
\begin{aligned}
\gamma &= 1 + 2\lambda \sum_{\delta=1}^{\infty} \frac{1}{(2\delta - 1)^{1+\sigma}}, \\
f(k) &= 2\lambda \sum_{\delta=1}^{\infty} \frac{\cos(2k\delta) - 1}{(2\delta)^{1+\sigma}}, \\
g(k) &= 2\lambda \sum_{\delta=1}^{\infty} \frac{\cos[(2\delta - 1)k]}{(2\delta - 1)^{1+\sigma}}.
\end{aligned}
\tag{38}
$$

This Hamiltonian is quadratic in creation and annihilation operators in quasimomenta and we intend to diagonalize the problem employing a Bogoliubov-Valatin transformation. Following Ref. [102], we introduce the operator

$$
\vec{\psi}^\dagger_k = \begin{pmatrix} \vec{c}^\dagger_k & \vec{c}^T_k \end{pmatrix} = \begin{pmatrix} a^\dagger_{k,1} & b^\dagger_{-k,1} & a^\dagger_{k,2} & b^\dagger_{-k,2} & a_{k,1} & b_{-k,1} & a_{k,2} & b_{-k,2} \end{pmatrix}.
\tag{39}
$$

We use this operator to bring the spin-wave Hamiltonian into canonical quadratic form

$$
\mathcal{H}^{\mathrm{SW}}_\parallel = \sum_k \left[ \frac{1}{2} \vec{\psi}^\dagger \underbrace{\begin{pmatrix} A_k & B_k \\ B^\dagger_k & A^T_k \end{pmatrix}}_{\equiv M_k} \vec{\psi} - \frac{1}{2} \mathrm{tr} A_k \right],
\tag{40}
$$

where $A_k$ and $M_k$ are Hermitian matrices and $B_k$ is a symmetric matrix. To solve the diagonalization problem we must find a transformation $\vec{\psi}_k = T \vec{\varphi}_k$ that brings $M_k$ into diagonal form and preserves the bosonic anticommutation relations of $\vec{\psi}_k$. Xiao [102] proofs that the problem can be reformulated in terms of the eigenvalue problem of the dynamic matrix

$$
D_k = \begin{pmatrix} A_k & B_k \\ -B^\dagger_k & -A^T_k \end{pmatrix},
\tag{41}
$$

arising from the Heisenberg equation of motion and that the transformation matrix $T$ can be constructed using appropriately normalized eigenvectors. A physical solution to the problem exists if and only if the dynamical matrix is diagonalizable and the eigenvalues are real. Employing this scheme we find

$$
\mathcal{H}^{\mathrm{SW}}_\parallel = \mathrm{const.} + S \sum_{k,\nu} \left( \omega_+(k) \alpha^\dagger_{k,\nu} \alpha_{k,\nu} + \omega_-(k) \beta^\dagger_{k,\nu} \beta_{k,\nu} \right),
\tag{42}
$$

in terms of the new boson creation and annihilation operators $\alpha_{k,\nu}^{(\dagger)}$ and $\beta_{k,\nu}^{(\dagger)}$ and the spin-wave dispersion

$$\omega_\pm(k) = \sqrt{(\gamma - f(k))^2 - (g(k) \pm 1)^2}. \tag{43}$$

In the limit $\lambda \to \infty$ we recover the spin-wave dispersion in Ref. [41] for the long-range Heisenberg spin chain. The staggered magnetization deep in the antiferromagnetic regime can be expressed as $m = S - \Delta m$ where $\Delta m$ is the correction induced by quantum fluctuations. We start with the expression

$$\Delta m = \sum_{\nu=1}^{2} \langle a_{j,\nu}^\dagger a_{j,\nu} \rangle \overset{N \to \infty}{=} \frac{1}{\pi} \sum_\nu^2 \int_{-\pi/2}^{\pi/2} dk \langle a_{k,\nu}^\dagger a_{k,\nu} \rangle, \tag{44}$$

and rewriting it in terms of the boson operators $\alpha_{k,\nu}^{(\dagger)}$ and $\beta_{k,\nu}^{(\dagger)}$ we find

$$\Delta m = \frac{1}{\pi} \int_{-\pi/2}^{\pi/2} dk \left[ \frac{1}{2} \left( \frac{\gamma - f(k)}{\omega_+(k)} + \frac{\gamma - f(k)}{\omega_-(k)} \right) - 1 \right]. \tag{45}$$

Introducing the linear Holstein-Primakoff transformation for the Hamiltonian $\mathcal{H}_{\bowtie}$ including diagonal long-range interactions the linear spin-wave Hamiltonian reads

$$\mathcal{H}_{\bowtie}^{\mathrm{SW}} = \mathrm{const.} + S \sum_k \left\{ \sum_\nu \left[ (\Gamma - f(k)) \left( a_{k,\nu}^\dagger a_{k,\nu} + b_{-k,\nu}^\dagger b_{-k,\nu} \right) + g(k) \left( a_{k,\nu} b_{-k,\nu} + a_{k,\nu}^\dagger b_{-k,\nu}^\dagger \right) \right] \right.$$
$$+ v(k) \left( a_{k,1} b_{-k,2} + a_{k,2} b_{-k,1} + a_{k,1}^\dagger b_{-k,2}^\dagger + a_{k,2}^\dagger b_{-k,1}^\dagger \right)$$
$$\left. + w(k) \left( a_{k,1}^\dagger a_{k,2} + a_{k,2}^\dagger a_{k,1} + b_{-k,1}^\dagger b_{-k,2} + b_{-k,2}^\dagger b_{-k,1} \right) \right\}, \tag{46}$$

where we introduced multiple prefactors defined as $\kappa = \kappa_1 + \kappa_2$, $\Gamma = \gamma + \kappa$ and as

$$\kappa_1 = 2\lambda \sum_{\delta=1}^{\infty} \frac{1}{((2\delta)^2 + 1)^{\frac{1+\sigma}{2}}},$$
$$\kappa_2 = 2\lambda \sum_{\delta=1}^{\infty} \frac{1}{((2\delta - 1)^2 + 1)^{\frac{1+\sigma}{2}}},$$
$$v(k) = 1 + 2\lambda \sum_{\delta=1}^{\infty} \frac{\cos(2\delta k)}{((2\delta)^2 + 1)^{\frac{1+\sigma}{2}}}, \tag{47}$$
$$w(k) = 2\lambda \sum_{\delta=1}^{\infty} \frac{\cos[(2\delta - 1)k]}{((2\delta - 1)^2 + 1)^{\frac{1+\sigma}{2}}}.$$

Again employing the same Bogoliubov-Valatin transformation we can derive the spin-wave dispersion

$$\omega_\pm(k) = \sqrt{[\Gamma - (f(k) \pm w(k))]^2 - [g(k) \pm v(k)]^2}, \tag{48}$$

and the corrections to the staggered magnetization

$$\Delta m = \frac{1}{\pi} \int_{-\pi/2}^{\pi/2} dk \left[ \frac{1}{2} \left( \frac{\Gamma - f(k) - w(k)}{\omega_+(k)} + \frac{\Gamma - f(k) + w(k)}{\omega_-(k)} \right) - 1 \right]. \tag{49}$$

For both Hamiltonians $\mathcal{H}_{\parallel}$ and $\mathcal{H}_{\bowtie}$ we evaluate the integrals $\Delta m$ numerically and use the consistency condition $\Delta m < S$ in the antiferromagnetic regime to approximate the quantum phase transition line.

# D  (Hyper-) scaling relations

In renormalization group (RG) theory the generalized homogeneity of the free energy density is exploited [103]. Connecting the critical exponents of observables with the derivatives of the free energy density and exploiting the homogeneity properties, the (hyper-) scaling relations

$$\gamma = (2-\eta)\nu\,, \qquad \text{(Fisher equality)} \tag{50}$$

$$\gamma = \beta(\delta-1)\,, \qquad \text{(Widom equality)} \tag{51}$$

$$2 = \alpha + 2\beta + \gamma\,, \qquad \text{(Essam-Fisher equality)} \tag{52}$$

$$2 - \alpha = (d+z)\,\nu \ \text{ for } \ d \le d_{\text{uc}}\,, \quad \text{(Hyperscaling relation)} \tag{53}$$

can be derived. However, the hyperscaling relation breaks down above the upper critical dimension due to dangerous irrelevant variables in the free energy sector since these variables cannot be set to zero as the free energy density becomes singular in this limit [104, 105]. Allowing the correlation sector to be affected by dangerous irrelevant variables for quantum systems in analogy to previous works in classical systems [106, 107] the hyperscaling relation can be generalized to

$$2 - \alpha = \left(\frac{d}{\digamma} + z\right)\nu\,, \tag{54}$$

with the pseudocritical exponent $\digamma = \max(1, d/d_{\text{uc}})$ [40]. As the one-dimensional $O(3)$ quantum rotor model can be mapped to the low-energy properties of the dimerized antiferromagnetic Heisenberg ladder [90] we can use the long-range mean-field critical exponents

$$\gamma = 1\,, \qquad \nu = \frac{1}{\sigma}\,, \qquad z = \frac{\sigma}{2}\,, \qquad \eta = 2-\sigma\,, \tag{55}$$

derived from one-loop RG [60] for the long-range $O(3)$ quantum rotor model at the upper critical dimension and insert them into Eq. (52). We find $d_{\text{uc}}(\sigma) = 3\sigma/2$. It directly follows that $d > d_{\text{uc}}$ in the regime $\sigma < 2/3$. Thus, we can rewrite

$$\digamma = \max\left(1, \frac{2}{3\sigma}\right) = \begin{cases} 1\,, & \text{for } \sigma \ge 2/3\,, \\ \frac{2}{3\sigma}\,, & \text{for } \sigma < 2/3\,, \end{cases} \tag{56}$$

which together with Eq. (54) is the generalized hyperscaling relation as derived in Ref. [40].

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
