# Peer review of "Continuously varying critical exponents in long-range quantum spin ladders"

_SciPost Physics, doi:SciPost Phys. 15, 087 (2023)_

## Round 1 · Referee Report · Nicolas Laflorencie (Referee 1) · 2023-4-14

Strengths

1- Interesting and well-posed physical problem
2- Results are quantitative and go beyond the state of the art
3- Technical details well balanced with the results and their interpretation

Weaknesses

1- Lack of experimental relevance/discussion
2- Some confusion between T=0 and finite T HMW theorem

Report

This paper addresses interesting quantum critical phenomena in long-range spin ladders using a combination of large-S (spin-wave) calculations and PCUT calculations. Overall the results are quite convincing, although not exact, as opposed to feasible (frustration-free) quantum Monte Carlo simulations. Nevertheless, the authors find that a deconfined criticality scenario is not present, contrasting with previous claims. The phase diagrams and estimates for critical exponents are obtained, and well presented.

Phase diagrams: it may have been useful to superimpose with models in order to ease their comparison. The SW value at infinite \lambda is not clearly explained (it ressembles a symbols)

Critical exponents: The plots do not show with enough clarity the departure from MF behaviors. The readability should be improved for Figs. 3 and 4. When comparing with Ref. [42] where \eta and z have been computed along the non-MF critical line, the behaviors are quite different. Indeed in the preprint the authors find \eta \le (2-\sigma) and z\ge \sigma/2. If one translates onto the notations of Ref. [42] where the regime \sigma\ge 1 was studied, it was found that \eta \ge 2-\sigma and z\le 1 (see Figs. 9 and 10 in [42]). Of course the nature of the transition here is different: the critical line separates LRO from a gapped state (in contrast with [42] where the transition is LRO -QLRO). It would be interesting to comment on this, in particular if such an observed behavior could be an artifact of the PCUT technique? Does it make sense to try to study this model with QMC?

In any case the present study is a really interesting work that clearly deserves publication.

I have a few additional minor comments to make.
First, I think the discussion about the Mermin-Wagner theorem is a bit misleading in the introduction. Mermin-Wagner does not say anything about T=0 LRO, but only about finite T (note that MW has been improved by Bruno, your Ref [70], stating that finite T LRO can occur if \sigma<1 ). The only work that I know addressing the T=0 case is [Néel order in the ground state of Heisenberg antiferromagnetic chains with long-range interactions, byRodrigo Parreira, O Bolina and J Fernando Perez, J. Phys. A 30, 1095 (1997)].

Talking about possible deconfined quantum criticality, probably better candidates would be frustrated models. How would PCUT perform if instead on the unfrustrated ladder, one would take for instance the J1-J2 two-leg ladder? See Phys. Rev. B 73, 214427 (2006) or Phys. Rev. B 84, 144407 (2011).

Finally, do you have in mind some possible experimental relevance for this setup? It would be interesting to motivate a bit more on this aspect.

Requested changes

1- The discussion about HMW theorem

2- Improvement on the readability of the critical exponents figures

  • validity: high
  • significance: high
  • originality: high
  • clarity: high
  • formatting: excellent
  • grammar: excellent

Author:  Patrick Adelhardt  on 2023-05-17  [id 3674]

(in reply to Report 1 by Nicolas Laflorencie on 2023-04-14)

We thank the referee for thoroughly reading the manuscript and the positive response recommending it for SciPost Physics. We will first comment on the points raised by the referee. Afterwards, we list the specific changes we made in the manuscript.

First, we like to address the raised question concerning the Hohenberg-Mermin-Wagner theorem. While we agree that the HMW theorem in the well-known formulations (Refs. 67-70) does not apply strictly to T=0, we want to point out the paper by Pitaevskii and Stringari from 1991 (Ref. 71), where the authors proof the absence of LRO for continuous symmetry groups in 1D at T=0. Additionally, we want to make the usual more hand-waving argument of quantum-classical mapping where a D-dimensional classical system can be mapped to a (d+1)-dimensional quantum system. So the 1D T=0 quantum case can be mapped to a 2D classical system where the HMW theorem applies.

Second, concerning the critical exponents, we thank the referee for explicitly pointing out the interesting difference in \eta and z between our preprint and Ref. 42. We do not believe that the behavior observed (\eta \le 2-\sigma and z>1) is an artifact of our pCUT approach. Sure, the estimates for the exponents become difficult to extract from the DlogPade approximants for large \sigma \lesssim \sigma^* as the critical point quickly shifts to larger values. However, in the intermediate regime between mean-field regime and \sigma^* the critical point is still < 0.75 and the series and DlogPadé approximants are well-behaved. There is also the possibility of additive or multiplicative corrections to the dominant power-law behavior, which potentially could spoil the critical exponents (e.g., the alpha-exponent about the upper critical dimension), yet it is hard to imagine a scenario where the true physical exponents deviate so much to show the "opposite" qualitative behavior. Knowing about the drawbacks of series expansions for \sigma->\sigma^*, we indeed just started a project to employ SSE QMC for this problem.

Third, concerning the frustrated spin ladders, we thank the referee for pointing out related physics in other spin-ladder models with frustrated but short-range interactions. One can certainly apply the pCUT method within the rung-singlet phase and study its quantum critical breakdown. We would leave this for an interesting future investigation.

Fourth, concerning a discussion about experimental realizations, we would first like to point out that a major motivation for our work is to further increase our understanding of quantum many-body systems with long-range interactions on a fundamental level. At the same time the experimental interest on such systems is steadily increasing due to the highly improved control of quantum-optical platforms. However, while the interactions in such systems are naturally long-range, most of the time the underlying microscopic Hamiltonians do not possess continuous symmetries like the Heisenberg system studied in our work. A notable exception is a recent work by the group of Immanuel Bloch, see Realizing the symmetry-protected Haldane phase in Fermi–Hubbard ladders, Pimonpan Sompet et al, Nature 606, 484 (2022). Here indeed a two-leg ladder system with SU(2) symmetry is realized with neutral atoms.

We have addressed the suggestions and comments by the referee in the revised manuscript as follows:

  • Referee: The plots do not show with enough clarity the departure from MF behaviors. The readability should be improved for Figs. 3 and 4.

Our answer: We thank the referee for this comment. We improved the readability of Figs. 3 and 4 by decreasing the line width of the data points and zooming in on the y-scale for some exponents (explicitly improving the readability of \eta and z).

  • Referee: When comparing with Ref. [42] where \eta and z have been computed along the non-MF critical line, the behaviors are quite different. Indeed in the preprint the authors find \eta \le (2-\sigma) and z\ge \sigma/2. If one translates onto the notations of Ref. [42] where the regime \sigma\ge 1 was studied, it was found that \eta \ge 2-\sigma and z\le 1 (see Figs. 9 and 10 in [42]). Of course the nature of the transition here is different: the critical line separates LRO from a gapped state (in contrast with [42] where the transition is LRO -QLRO).

Our answer: We have added a sentence on page 9 before the conclusions stating this interesting difference between the two models as well as the difference to the field-theoretic expectation. We refer explicitly to Ref. [42].

  • Referee: Phase diagrams: it may have been useful to superimpose with models in order to ease their comparison. The SW value at infinite \lambda is not clearly explained (it resembles a symbols)

Our answer: We thank the referee for this suggestion. However, we would not like to superimpose both phase diagrams, because then easily the impression arises that the phase diagram contains an "intermediate phase" (between the phase transition lines of the models) which would be irritating. Concerning the second issue, we fully agree with the referee and we have erased the symbols for the limiting SW case.

  • Referee: I think the discussion about the Mermin-Wagner theorem is a bit misleading in the introduction. Mermin-Wagner does not say anything about T=0 LRO, but only about finite T (note that MW has been improved by Bruno, your Ref [70], stating that finite T LRO can occur if \sigma<1 ). The only work that I know addressing the T=0 case is [Néel order in the ground state of Heisenberg antiferromagnetic chains with long-range interactions, byRodrigo Parreira, O Bolina and J Fernando Perez, J. Phys. A 30, 1095 (1997)].

Our answer: As stated above, reference [71] shows the absence of LRO (breaking of continuous symmetries) at T=0 for one-dimensional quantum systems. However, we acknowledge the confusion about the finite T and zero T cases and modified the corresponding sentence in the draft accordingly.

  • Referee: Finally, do you have in mind some possible experimental relevance for this setup? It would be interesting to motivate a bit more on this aspect.

Our answer: We added the reference Nature 606, 484 (2022) to the introduction and noted that the symmetry-protected phases (exhibiting non-local string order) were experimentally realized on ladder geometries.

---

## Round 1 · Referee Report · Anonymous (Referee 2) · 2023-4-16

Strengths

1) Considers a class of systems of growing importance

2) Reports valid results for critical exponents

Weaknesses

1) Nothing very striking or surprising is reported

Report

There is a growing interest in quantum spin models with long-range
(power-law decaying) interactions - for both experimental and more
theoretical reasons. This paper presents a useful contribution in
this area, using high-order expansions around the rung limit of a
ladder system.

The main aim here is to compute critical exponents of the system
at the transition between the ordered Neel phase and the rung
singlet phase. The exponents vary with the interaction parameters
and results that appear stable/reliable are obtained. The work
is carried out carefully, even considering the log corrections
when the system is effectively at the upper critical dimension.

I do not think any of the results are particularly striking or
surprising, but overall they are certainly useful. One of the
main conclusions is that a previous claim of deconfined criticality
[71] does not seem to hold up. If I understand correctly, this
conclusion is based on the result that the dynamic exponent is
not one. Here I note that the true nature of the deconfined
criticality has come under renewed scrutiny, and it is not
completely clear what transitions should even fall under this
umbrella - perhaps z=1 is not even required and there are some
works showing transitions that are at least in some way related
to deconfined criticality but where Lorentz invariance is broken
(e.g., PRB 104, 045110 (2021) and previous papers where a spin
liquid connected to a dqc point are discussed). Discussing
these recent developments may go a bit too far from the topic
of the paper but I wanted to point it out for consideration
by the authors in case there is something else in their
results that could support (or not) the deconfined criticality
scenario beyond z not being one. The authors do point out
that the rung singlet phase is trivial, which in itself seems
to suggest that no unusual mechanisms need to be invoked.

The paper can be published after the authors have considered
the above suggestion (any changes are optional). I recommend that
the authors also check the English one more time; while the paper
overall reads very well, there are some minor errors, e.g., the
line after Eq. (2), where "distant-dependence" should be
"distance-dependence".

  • validity: high
  • significance: good
  • originality: good
  • clarity: high
  • formatting: excellent
  • grammar: excellent

Author:  Patrick Adelhardt  on 2023-05-17  [id 3675]

(in reply to Report 2 on 2023-04-16)

We thank the referee for his/her thorough reading of the manuscript and recommendation for publication.

Referee: One of the main conclusions is that a previous claim of deconfined criticality [72] does not seem to hold up. If I understand correctly, this conclusion is based on the result that the dynamic exponent is not one. Here I note that the true nature of the deconfined criticality has come under renewed scrutiny, and it is not completely clear what transitions should even fall under this umbrella - perhaps z=1 is not even required and there are some works showing transitions that are at least in some way related to deconfined criticality but where Lorentz invariance is broken (e.g., PRB 104, 045110 (2021) and previous papers where a spin liquid connected to a dqc point are discussed). Discussing these recent developments may go a bit too far from the topic of the paper but I wanted to point it out for consideration by the authors in case there is something else in their results that could support (or not) the deconfined criticality scenario beyond z not being one.

Our answer: We thank the referee for this comment. Let us first stress that z=1 is not the only finding which is in conflict with the scenario outlined in [72], but also the fact that we observe continuously varying critical exponent inline with field-theoretical expectations for the O(3) quantum rotor model. As the referee pointed out, discussing the recent developments he/she outlined may go a bit too far from the topic of the paper. We therefore would stick to the discussion in the current form of the manuscript.

Referee: I recommend that the authors also check the English one more time; while the paper
overall reads very well, there are some minor errors, e.g., the line after Eq. (2), where "distant-dependence" should be
"distance-dependence".

Our answer: We thank the referee for pointing out the typos. We have corrected them in the revised version.

---

## Round 2 · Referee Report · Anonymous (Referee 3) · 2023-5-17

Report

The authors have significantly improved the manuscript by addressing the issues brought up by the referees. It can now be published.

---

## Round 2 · List of Changes

l. 50-51: Added experimentally relevant reference. We emphasize that symmetry-protected topological phases were detected on ladder geometries.

l. 63: Removed the word 'here'.

l. 67-68: We split up references for HMW therom for the finite and zero temperature case.

l. 92: Spelling mistake corrected: distant-dependent -> distance-dependent

l. 130: for->to

l. 143: Added minus sign to formula

l. 168: Added 'to' to sentence

l. 173-174: Updated references in figure 2 and removed SW symbols at \lambda=\infty

l. 197-198: Improved readability of figure 3.

l. 206-207: Improved readability of figure 4

l. 223-225: We added a sentence comparing z and nu to Ref. 43.

l. 272: Added an url to Zenodo, providing the raw data of our simulations.

l. 370-371: Added minus sign to formula

l. 386: Added 'so' to sentence

l. 387: Changed S to \mathcal S in formula

---

## Editorial Decision

published